# Micro-Laser-Induced Breakdown Spectroscopy: A Novel Approach Used in the Detection of Six Rare Earths and One Transition Metal

**Madhavi Martin [1],\*, Daniel Hamm [2], Samir Martin [3], Steve Allman [1], Gary Bell [2] and Rodger Martin [2]**

[1]   Biological Sciences Division, Oak Ridge National Laboratory, Oak Ridge, TN 37831, USA; allmansl@gmail.com

[2]   Fusion and Materials for Nuclear Systems Division, Oak Ridge National Laboratory, Oak Ridge, TN 37831, USA; dhamm@vols.utk.edu (D.H.); bellgl@ornl.gov (G.B.); martinrc@ornl.gov (R.M.)

[3]   Department of Cardiology, Emory School of Medicine, Emory University, Atlanta, GA 30322, USA; Samir.martin@emory.edu

\*   Correspondence: martinm1@ornl.gov; Tel.: +1-865-574-7828

**Abstract:** Laser-induced breakdown spectroscopy (LIBS) was undertaken using an instrument which used a high-powered microscope to deliver the light and tightly focused the low energy laser beam onto the surface of a solid sample. A micro-plasma was generated on the surface of the sample under test even though the amount of energy/pulse from a beam of 532 nm was <1 mJ. Rare earth elements such as europium, gadolinium, lanthanum, neodymium, praseodymium, samarium, and a transition metal, yttrium, were tested. These elements are important in nuclear fission reactions especially for estimation of actinide masses for non-proliferation "safeguards". Each element was mixed in the graphite matrix in different percentages from 1% to 50% by weight and the LIBS spectra were obtained for each composition as well as after mixing each element in the same amount using oxides of the elements. The data for the 5% mixture of the rare earth elements with graphite powder along with the transition metal has been presented in this article. A micro-LIBS approach was used to demonstrate that these rare earth elements can be identified individually and in a complex mixture in glove boxes in which the microscope LIBS instrument is housed in a nuclear research environment.

**Keywords:** micro-laser-induced breakdown spectroscopy; rare earth elements; elemental peaks detection; micro-plasma

## 1. Introduction

Laser-induced breakdown spectroscopy (LIBS) analysis of nuclear materials is realizing increased interest for actinide mass and isotopic measurements. LIBS offers advantages over conventional solution-based radiochemistry in terms of cost, analytical turnaround, waste generation, personnel dose, and contamination risk [1–6]. For example, conventional analysis can require million-fold dilutions of high-activity samples, complicating impurity analyses. This article will provide an insider's look at the challenges and potential for routine LIBS application to high-level radiological samples. By identifying practical needs in non-routine sample analysis, LIBS can supplement conventional methods by providing rapid sample characterization of solid and concentrated liquid samples. As a microanalytical (submicrogram) sampling technique, LIBS can provide analysis of the limited sample masses permitted for high-level materials outside radiological hot cells. For isotopes such as Pu-238 or Cm-244, glove boxes are typically limited to subgram quantities, and chemical hoods to submicrograms [3,5]. The rare earths have been chosen to be studied based on the information

provided in Reference [7], which reports that in a nuclear fission reaction, ~18% of fissions produce Nd atoms, 13% Ce atoms (previously studied [8]), 6% La and Pr, 4% Y and Sm, 0.6% Eu, and 0.3% Gd. These same elements have been studied previously to obtain quantitative analysis using PLS (partial least square) technique in a prior publication by the same authors [8]. A number of articles have been published describing the detection and monitoring of europium oxides and $Eu^{2+}$ in colloids [9–14]. LIBS for the detection of gadolinium in molten glass, in coated stainless steel plates, and in its oxide form have been reported in References [15–17]. Lanthanum in molten glass and alloys of neodymium have also been reported [15,18–22]. Studies to detect samarium as a trace pollutant in soils, and its alloys using the LIBS technique have been published [23,24]. Yttrium as a component of iron garnets and as a toxic metal in incinerator stack exhausts has been reported in References [25–27]. LIBS has also been used to detect a large number of lanthanides in raw monazite sands [28]. Some statistical analysis is needed to develop calibration and validation models [29,30] for the LIBS work done previously.

The technique of choice for measuring the actinide and rare earth content in irradiated fuel is ICP-MS (Inductively Coupled Plasma-Mass Spectrometry) or ICP-OES/AES (Inductively Coupled Plasma-Optical Emission Spectroscopy/Atomic Emission Spectroscopy) [31–35], but the challenge of analyzing concentrated solutions from fuel dissolution with very high radiological dose must first be addressed. This creates analytical uncertainties from large dilutions, up to a million-fold, which can be costly and time consuming and increase worker hazards in sample handling. Furthermore, complex spectral features can create problems in identification, fingerprinting, and in the quantification of these elements.

In this journal article, the use of micro-LIBS (Laser delivered to a sample via a microscope objective) has been studied in detail for the identification of the rare earth elements mentioned previously. The LIBS detection optical system was coupled to a laser scribing system in a very elegant configuration that is discussed in this article.

## 2. Materials and Methods

### 2.1. Laser Scribing System

This research uses a commercial Quiklaze 200 mJ Nd:YAG laser scribing system (custom made), ESL, Elemental Scientific Lasers, Bozeman, MT, shown in Figure 1a,b, and has been modified to perform laser-induced breakdown spectroscopy. A pulsed laser is set on top of a microscope and the light is focused by 50X and 100X objective lenses onto the surface of a sample placed at the focus of the lens. The spot size generated by the 50X objective lens is 50 μm and with a 100 X objective lens the spot size obtained is 10 μm. This laser system produces a 1064 nm (IR) wavelength laser pulse, which can be frequency doubled and tripled to produce 532 nm (Green) and 355 nm (UV) laser pulses. For this research, only the 532 nm wavelength light was used. The New Wave Quicklaze system custom made) ESL, Elemental Scientific Lasers, Bozeman, MT) is equipped with an automated 3-dimensional stage and a camera system. This allows for *X-Y-Z* movement while sampling in addition to being able to see the sample surface and the etching/scribing of the sample surface via a camera assembly. Now, in addition to these existing features, the ability to obtain the chemical composition of the sample via the collection of optical emission from the sample has been added to this system.

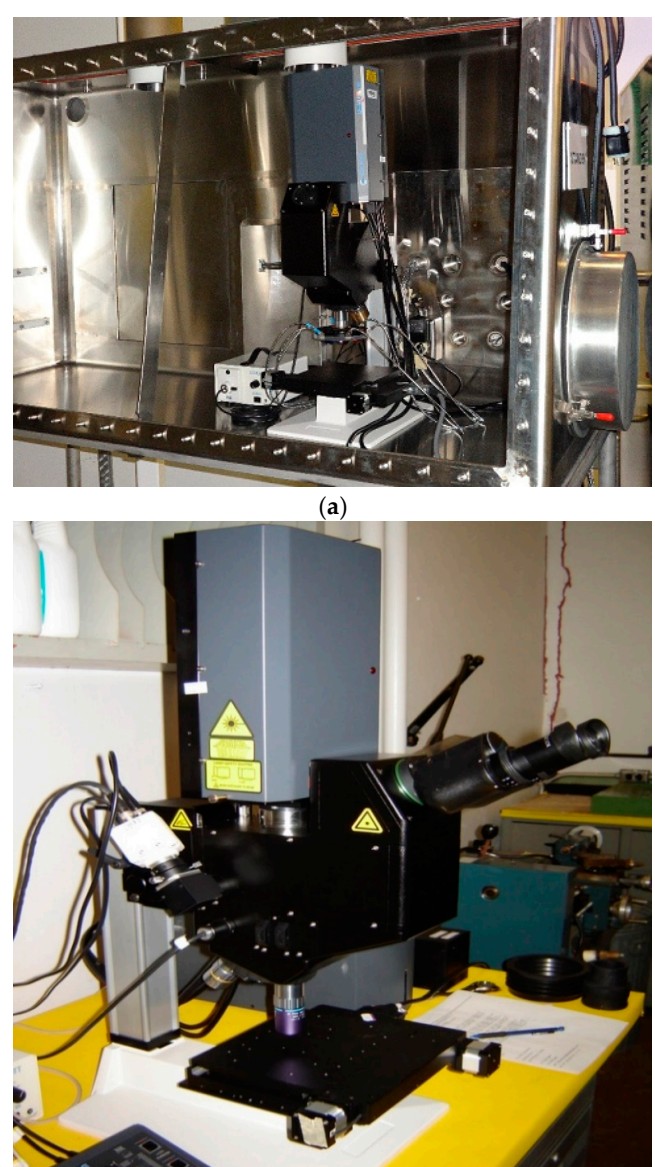

(a)

(b)

**Figure 1.** Quiklaze 200 laser scriber customized and coupled to a LIBS optical system: (**a**) The Quiklaze 200 laser scribe placed in a glovebox and (**b**) Quiklaze 200 laser scriber coupled to a LIBS optical collection assembly.

The scribing system is designed to inscribe insignias and identifying markers onto samples at the micrometer scale and has been customized for implementation in a glove box environment as well. Its original inscription purposes included use on radioactive and contaminated materials to identify these samples in a nuclear reactor environment. The samples that are inserted into a reactor to be irradiated need to be inscribed. The inscriptions on the samples are used to understand the placement sequence in the reactor where they are irradiated. The laser scribing system can also be used for cutting on a microscopic level, or to remove metal shorts that are created during microelectronics circuit fabrication. The laser beam in this scriber is designed to be guided along the optical path within the body of the microscope and is emitted at the output objective.

*2.2. LIBS Detection Coupled to the Laser Scribing System*

For LIBS, the Quiklaze 200 is equipped with an optical collection assembly, shown in Figure 2, consisting of six optical emission collectors and six fiber optic cables. The spectra acquired using the

detection module provided by Applied Photonics Ltd were taken using single shots. The laser has a wavelength of 532 nm with a pulse length nominally between 4–6 ns. The LIBS detection module added to the scribing system utilizes six Avantes (Avantes BV, Apeldoorn, The Netherlands) spectrometers that cover the wavelength range of 182–904 nm and the CCD (Charge Coupled Device/Detector) arrays are operated with a gate width of 1.1 milliseconds and with a delay of 1.27 microseconds. The spectral resolution is different for the different spectrometer channels. Specifically, the resolution for Channel 1, covering 182–256 nm, Channel 2, covering 255–315 nm and Channel 3, with wavelength range of 314–416 nm all have the FWHM (Full Width at Half Max) = approx. 0.06 nm. In case of Channel 4 which covers the wavelength range of 414–498 nm has a FWHM of 0.08 nm. The last two channels, Channel 5 (496–718 nm) and Channel 6 (716–904 nm) both have a FWHM of 0.18 nm, Applied Photonics Ltd (Skipton, UK) has provided the ability for optical collection of the emission from the spark that is generated on the sample surface and delivery to a bank of CCD array spectrometers and detector system. This provides the dual-use capability for the modified instrument and allows for in situ qualitative identification of materials in principle [8]. This report seeks to validate these capabilities and prove its usefulness as a spectroscopic instrument. Figure 2 shows the AutoCAD design of the optical collection system (a) side view and (b) close-up from the bottom. This attachment has added the benefit of being able to do elemental detection and analysis along with the original purpose of using the system for making patterns on a sample surface which would identify samples before and after these samples were irradiated in a nuclear reactor.

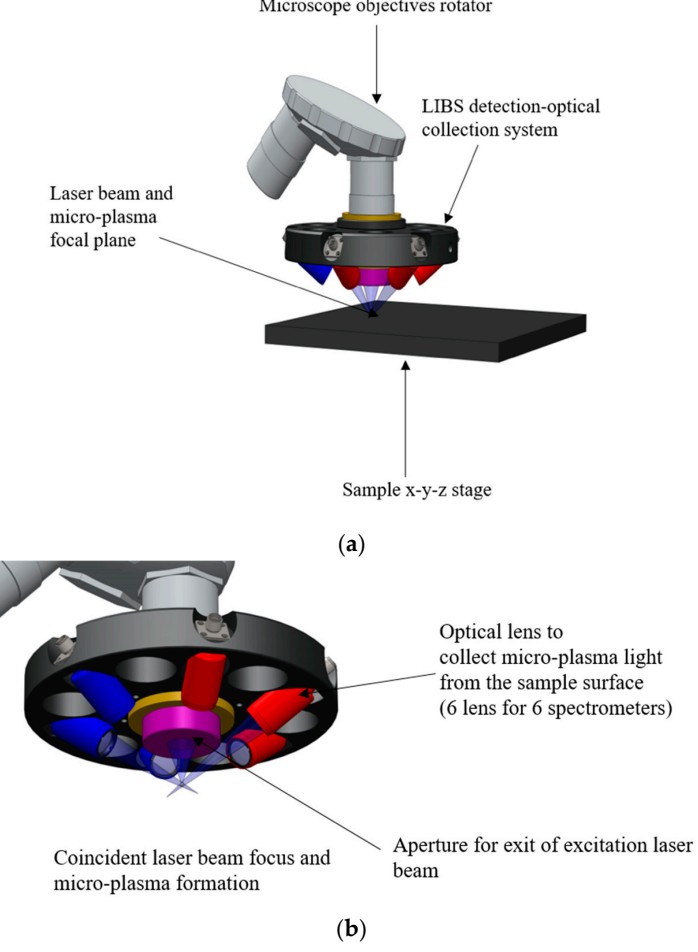

**Figure 2.** AutoCAD depiction of the optical collection system: (**a**) side view and (**b**) close-up from the bottom.

### 2.3. Materials and Preparation

Excess material of the rare earth powders was obtained from other researchers in the organization. The powders of the rare earth oxides that were obtained were europium oxide ($Eu_2O_3$), gadolinium oxide ($Gd_2O_3$), lanthanum oxide ($La_2O_3$), neodymium oxide ($Nd_2O_3$), praseodymium oxide ($Pr_6O_{11}$), samarium oxide ($Sm_2O_3$), and the transition metal oxide, yttrium oxide ($Y_2O_3$). The purity of the oxide powders that were used in this research was 99.99%. The graphite powder that was a natural, microcrystal grade, of 99.9995% purity was also obtained from Alpha Aesar (Ward Hill, MA, USA). The polyvinyl acetate (PVA) was 99–100% hydrolyzed and was obtained from Acros Organics (Waltham, NJ, USA). These powders were weighed out, and a balancing amount of graphite powder was also weighed and combined in a bottle. The mixture was vortexed and stirred. 300 μL of 0.5% PVA was pipetted into a glass tube and re-mixed a second time as in the previous step. The PVA was mixed with the powders and then dried on a heating block. The contents of the glass tube were emptied into a 1/4" die, pressed at 1500 1b for one minute. The pellets were placed in a labeled plastic bag and LIBS measurements were performed on them.

### 3. Results and Discussion

The plasma created by a 532 nm laser pulse of 0.5 mJ energy is very faint or weak, making it difficult to obtain the emission spectra from the sample under test. It was hypothesized that if the spark can be seen by the human eye then the bank of the CCD detector array should be able to detect the emission spectrum for any solid sample that is being tested. This hypothesis was proven to be true. The spectra for all the rare earth listed above with a mix of 5% rare earth and 95% graphite pellets were obtained. The data for the 5% mixture of the rare earth elements with graphite powder along with the transition metal has been presented here in Figure 3. The objective used for the collection of the emission peaks for all of the rare earth elements is identified to be 50X. Figure 3 shows the spectra for them.

Figure 3a shows the emission spectra for the rare earth Eu. The main spectral features such as, 381.967 nm (II), 393.048 nm (II), 397.196 nm (II), 420.505 nm (II), 443.556 nm (II), 452.257 nm (II), 459.403 nm (I), 462.722 nm (I), 466.188 nm (I), 490.086 nm (I), 535.761 nm (I), 548.865 nm (I), 576.520 nm (I), 583.098 nm (I), and 596.607 nm (II) have been detected. This shows that 5% of the europium and other rare earth elements in 95% of the graphite matrix can be detected quite easily. The successful acquisition of micro-LIBS spectra for all the rare earth elements was performed and shown in Figure 3a–g. The authors have further identified the emission peaks for these samples that are detected in the UV-region of the wavelength region (190–450 nm). More detailed spectral features have been shown and discussed in another publication [8].

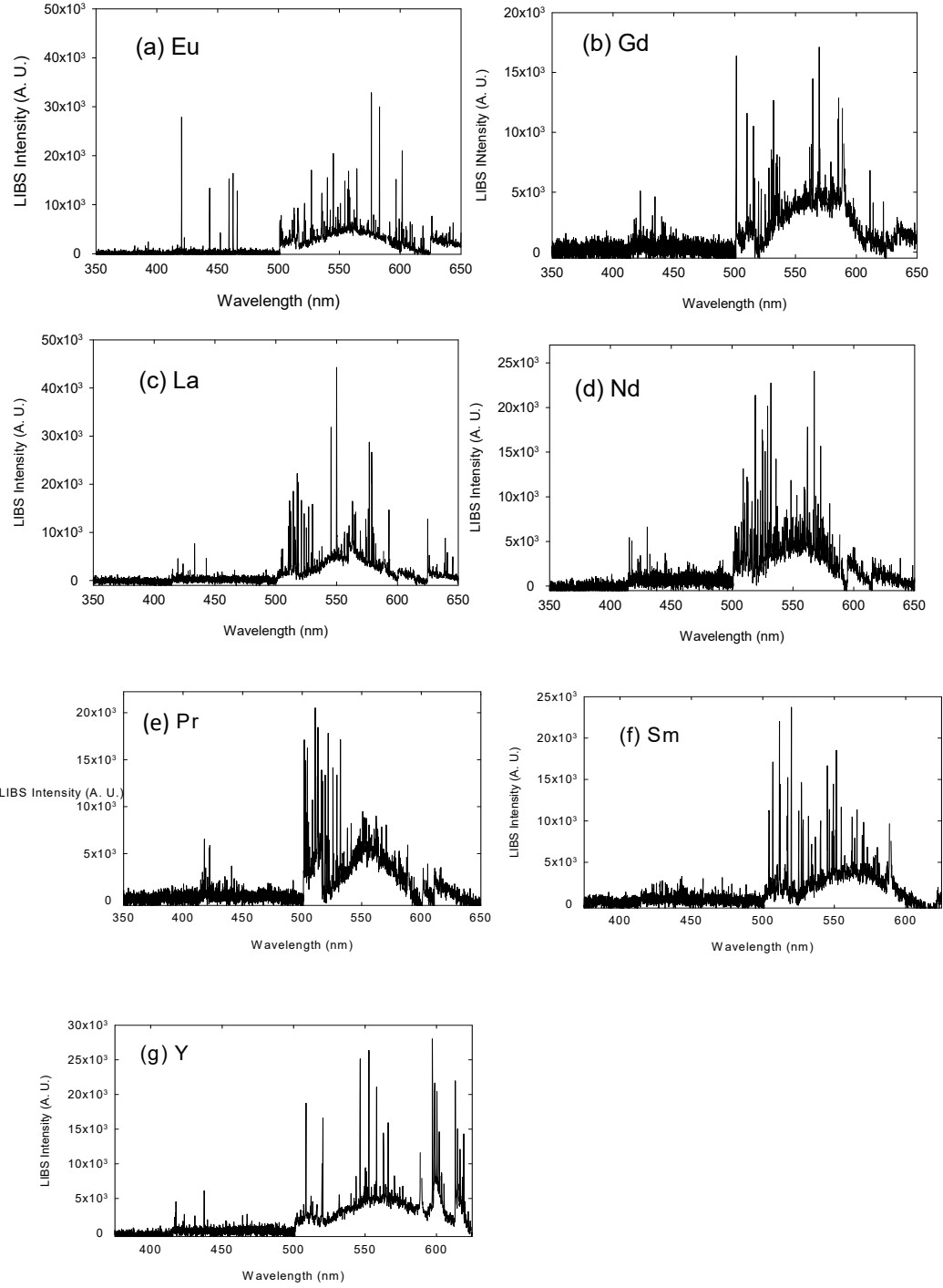

**Figure 3.** The broadband LIBS spectra for 5% of (**a**) europium, (**b**) gadolinium, (**c**) lanthanum, and (**d**) neodymium, (**e**) praseodymium, (**f**) samarium, and (**g**) yttrium in 95% graphite powder.

The distinguishing features for all the elements that were studied are shown in Table 1.

**Table 1.** Major emission wavelengths contributing to the LIBS spectra for each of the rare earth elements.

| Element | Wavelength (nm) | Element | Wavelength (nm) |
|---|---|---|---|
| Eu II | 381.967 | La II | 415.197 |
| Eu II | 393.048 | La I | 418.732 |
| Eu II | 397.196 | La II | 419.655 |
| Eu II | 420.505 | La II | 423.838 |
| Eu II | 443.556 | La II | 429.605 |
| Eu II | 452.257 | La II | 433.374 |
| Eu I | 459.403 | La II | 452.237 |
| Eu I | 462.722 | La II | 452.612 |
| Eu I | 466.188 | La II | 455.846 |
| Eu I | 490.086 | La I | 521.186 |
| Eu I | 535.761 | La I | 545. 515 |
| Eu I | 548.865 | La I | 639.423 |
| Eu I | 576.520 | Pr II | 417.939 |
| Eu I | 583.098 | Pr II | 422.293 |
| Eu I | 596.710 | Pr II | 422.535 |
| Nd II | 415.626 | Pr II | 428.242 |
| Nd II | 417.732 | Pr II | 440.882 |
| Nd II | 432.576 | Pr II | 446.866 |
| Nd II | 433.88 | Pr I | 473.669 |
| Nd II | 445.157 | Pr I | 492.460 |
| Nd II | 527.343 | Pr II | 511.038 |
| Nd II | 543.153 | Pr I | 513.344 |
| Nd II | 582.587 | Pr II | 522.011 |
| Gd I | 422.585 | Pr II | 532.276 |
| Gd I | 432.712 | Pr II | 535.240 |
| Gd I | 501.504 | Pr I | 552.415 |
| Gd I | 510.345 | Pr I | 553.837 |
| Gd I | 515.584 | Pr II | 562.305 |
| Gd I | 537.063 | Pr II | 581.533. |
| Gd I | 545.346 | Sm I | 429.674 |
| Gd I | 561.791 | Sm II | 443. 388 |
| Gd I | 570.135 | Sm I | 471.610 |
| Gd I | 574.636 | Sm I | 488.377 |
| Gd I | 577.602 | Sm I | 511.716 |
| Gd I | 579.138 | Y II | 430.963 |
| Gd I | 585.163 | Y II | 437.494 |
| Gd II | 585.524 | Y II | 439.802 |
| Gd I | 585.622 | Y II | 508.742 |
| | | Y I | 570.671 |

The detailed identification of the emission peaks in the broadband spectra is very difficult to label in Figure 3a–g. The individual enhanced spectra for (a) europium, (b) gadolinium, (c) lanthanum, and (d) neodymium are shown below in Figure 4a–d. These spectra show the specific micro-LIBS emissions lines in a smaller wavelength range for each of the elements mentioned above. The six Avantes spectrometers that are used to detect the plasma emission from the sample being tested do not have the same optical efficiency throughout the whole wavelength range of 195–904 nm. This efficiency is very low for the UV-VIS (Ultra Violet-Visible) part of the broadband region of spectrometers (specifically from 195–450 nm). This is demonstrated in the broadband spectra for each of the rare earth elements and the transition metal. The majority of the emission lines are observed in the wavelength range of 350–600 nm (Figure 3). The above spectra in Figure 3a–g have been enhanced to show micro-plasma that was detected by the spectrometers even in the low optical efficiency wavelength range.

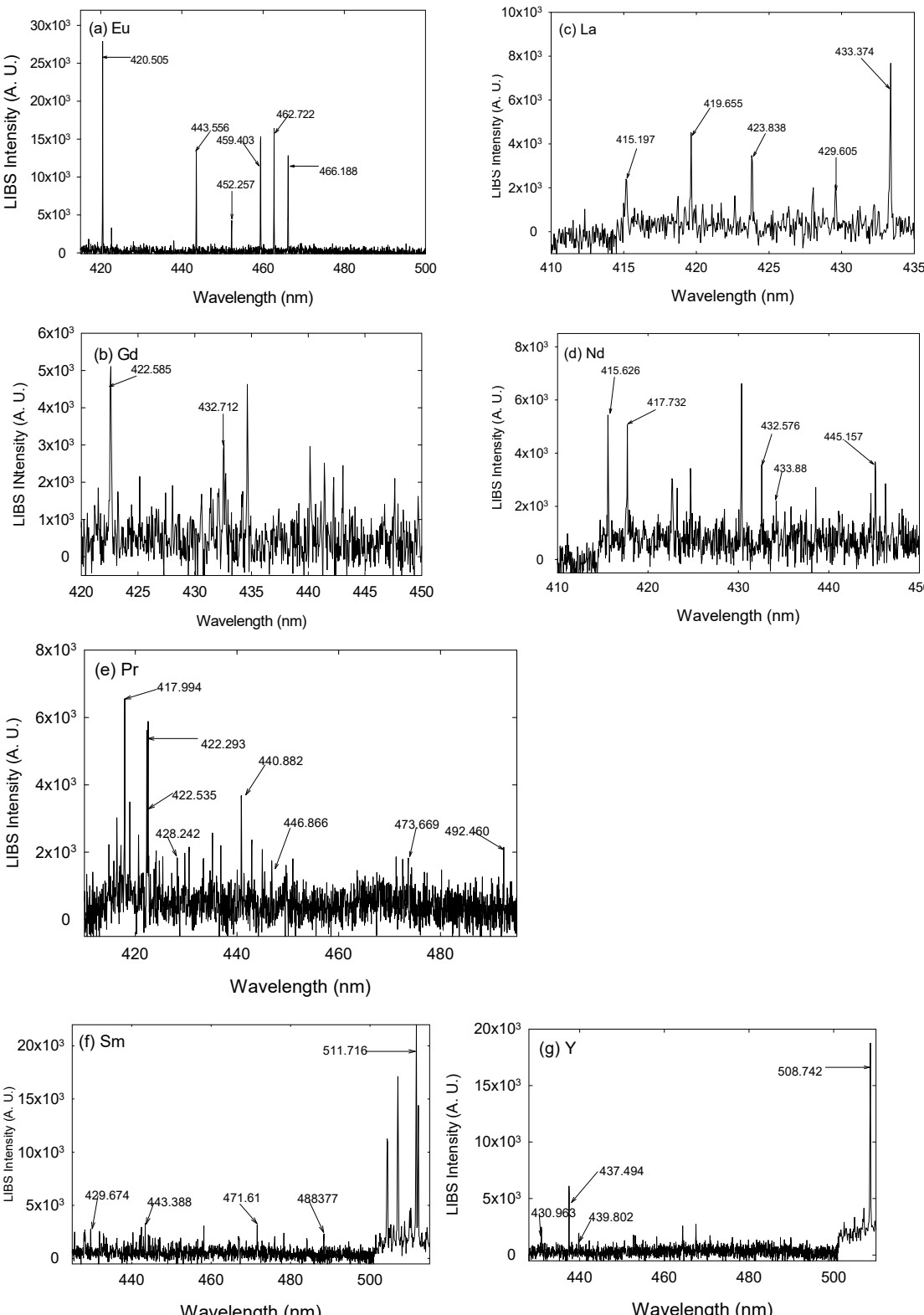

**Figure 4.** The LIBS spectra for 5% of (**a**) europium, (**b**) gadolinium, (**c**) lanthanum, and (**d**) neodymium (**e**) praseodymium, (**f**) samarium, and (**g**) yttrium in 95% graphite powder in a narrow wavelength range showing the individual peaks that are characteristics of the elements shown here.

Similarly, Figure 4e–g emphasize the LIBS spectra for 5% of (e) praseodymium, (f) samarium, and (g) yttrium, in 95% graphite powder in a narrow wavelength range showing the individual peaks that are characteristic of the elements labeled here.

The LIBS spectra of all the rare earth elements separately have been obtained using a laser beam guided through a microscope coupled to a LIBS collection system for the first time. A nanosecond laser pulse with an energy/pulse of less than a millijoule has been used successfully to fingerprint all of these elements. Numerous peaks which are representative of the different rare earth elements and the one transition metal were identified.

## 4. Conclusions

The successful detection of rare earth elements such as Eu, Gd, La, Nd, Pr, Sm, and one transition metal, Y, was performed using a LIBS optical detection system coupled with a laser scribing instrument for the first time. The rare earth oxides and transition metal oxides were mixed in a 5% oxide and a 95% graphite matrix. All of the samples that were tested using the LIBS technique showed prominent peaks of the rare earth elements and of the transition metal in the graphite matrix. A microscope was used to focus the laser onto the surface of the sample and a very complex, but elegant, design for the collection of the plasma emission for the spectral acquisition was performed. A very innovative collection optics configuration was used to collect light from the micro-plasma that was generated by the laser-guided through the body of the microscope and focused at the exit via a microscopic lens onto the sample surface. This was achieved for the first time to identify and fingerprint all of these rare-earths and the transition metal that were tested. In the future, the same experimental configuration will be used to quantify and measure the limits of detection of the elements that were tested.

**Author Contributions:** M.M. conceptualized the experiment and the experimental setup which had to be customized and modified for dual usage for the purpose of scribing materials and detection of optical emission from plasma generated on the surface of the samples. Additionally, M.M. prepared the original draft of manuscript, and data analysis. D.H. did all the experiments, acquired the LIBS data, identified the emission peaks for all the rare earth elements, and edited the article. S.M. contributed in extensive editing of the manuscript and formatting of the manuscript. S.A. contributed in tedious sample preparation of multiple concentration for each rare earth elements and making pellets for those. G.B. instrumental in funding acquisition and involved in data acquisition, supervision, and project administration. R.M. helped with the writing, reviewing, experimental supervision, and project administration.

**Funding:** This manuscript has been authored by UT-Battelle, LLC under Contract No. DE-AC05-00OR22725 with the U.S. Department of Energy. The United States Government retains and the publisher, by accepting the article for publication, acknowledges that the United States Government retains a non-exclusive, paid-up, irrevocable, worldwide license to publish or reproduce the published form of this manuscript, or allow others to do so, for United States Government purposes. The Department of Energy will provide public access to these results of federally sponsored research in accordance with the DOE Public Access Plan (http://energy.gov/downloads/doe-public-access-plan).

**Acknowledgments:** This research was supported by the U.S. Department of Energy Office of nuclear materials program. This manuscript has been authored by UT-Battelle LLC under Contract No. DE-AC05-00OR22725 with the U.S. Department of Energy. The United States Government retains and the publisher, by accepting the article for publication, acknowledges that the United States Government retains a non-exclusive, paid-up, irrevocable, worldwide license to publish or reproduce the published form of this manuscript, or allow others to do so, for United States Government purposes.

**Conflicts of Interest:** The authors declare no conflict of interest.

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
