# Peer review of "Micro-Laser-Induced Breakdown Spectroscopy: A Novel Approach Used in the Detection of Six Rare Earths and One Transition Metal"

_minerals, doi:10.3390/min9020103_

Round 1

Reviewer 1 Report

The authors have proposed a very interesting experimental set-up for the detection of rare earth elements by micro LIBS measurements. The issue is well adherent to the aims of the journal since in the manuscript an advanced analytical technique is proposed.

I have some points to rise

1)      Experimental section: in my opinion the description of the scribing system could be left out, since it is not very useful for LIBS experiments. On the other hand, laser pulse duration is a key parameter and should be reported.

2)      Spectral resolution obtained in the experimental conditions should be reported.

3)      In all spectra shown in fig.3 a broad band centered at about 570nm is clearly visible. Have the authors any hypothesis about its origin?

4)      Authors have analyzed samples with only one of the selected elements at a time. Which is their opinion about the capability of the proposed experimental set-up to discriminate among several spectral lines in a multielemental sample?

Some typing errors are present ( line 188)

Author Response

Thanks to Reviewer #1.

1)      Experimental section: in my opinion the description of the scribing system could be left out, since it is not very useful for LIBS experiments. On the other hand, laser pulse duration is a key parameter and should be reported.

Response: We think that since this is the first time that we have integrated the scriber with the LIBS experiment, it is important to give a thorough description of this instrumentation in this manuscript.  Also we have added the laser pulse duration in the body of the manuscript.

2)      Spectral resolution obtained in the experimental conditions should be reported

Response: The spectral resolution and other experimental conditions have been added to section 2.2.

3)      In all spectra shown in fig.3 a broad band centered at about 570nm is clearly visible. Have the authors any hypothesis about its origin?

Response: Yes this has been explained on pg. 7, lines 170-177.

4)      Authors have analyzed samples with only one of the selected elements at a time. Which is their opinion about the capability of the proposed experimental set-up to discriminate among several spectral lines in a multielemental sample?

Response: This is a very good point which will be the subject for the next publication.  Ans has been mentioned in the conclusion section.

Some typing errors are present ( line 188)

This has been corrected.

Reviewer 2 Report

LIBS is used as a new application of a microanalytical (submicrogram) sampling technique for monitoring important elements produced by nuclear fission reactions ( detection of rare earth elements Eu, Gd, La, Nd, Pr, Sm, and one transition metal). These elements are important in nuclear fission reactions especially for estimation of actinide masses. This article would be interested for those who research nuclear decay products and safety of nuclear fuel usage.

Author Response

Thanks to Reviewer #2.

Reviewer 3 Report

Dear Authors

The development of a LIBS instrument starting from Laser Scribing System coupled with an optical collection system results a smart solution to perform in house chemical analyses.

The approach that you used has been already presented in another paper not cited (M. Martin, R. C. Martin, S. Allman, D. Brice, A. Wymore, N. Andre, 2015, Quantification of rare earth elements using laser-induced breakdown spectroscopy, Spectrochimica Acta, Part B, 114, 65–73).

I suggest some corrections:

check the numeration of each spectrum that compose figures 3 and 4.

In figures 3 and 4 the wavelength range should be the same for all the elements identified.

All intense lines should be identified

There are other elements in the graphite matrix? Can you identify the C emission lines?

The application of this method to the study of minerals would be very useful and advantageous, even if in your paper it is not clear what is the detection limit for the rare earth elements in the more complex matrix. The hope is that you can define this in a next article.

Author Response

Thanks to Reviewer #3.

1. The approach that you used has been already presented in another paper not cited (M. Martin, R. C. Martin, S. Allman, D. Brice, A. Wymore, N. Andre, 2015, Quantification of rare earth elements using laser-induced breakdown spectroscopy, Spectrochimica Acta, Part B, 114, 65–73).

Response : We have added this citation as reference # 28.

2. check the numeration of each spectrum that compose figures 3 and 4.

In figures 3 and 4 the wavelength range should be the same for all the elements identified.

Response: We have checked the numbering of the figures 3and 4.  The wavelength range has been kept different for the different elements because the peaks for each of these elements are present in different ranges of the broad band spectrum and so we are zooming into the regions where majority of the peaks are situated. 

3. All intense lines should be identified

There are other elements in the graphite matrix? Can you identify the C emission lines?

Response:  Most of the intense lines have been identified.  There are a few that are not available to us.

The graphite matrix has very low impurities of other elements and we have not been able to see any with our technique. They seem to be below the detection limit of LIBS. The carbon line at 247.85 nm has been observed but not shown in our graphs.

4. The application of this method to the study of minerals would be very useful and advantageous, even if in your paper it is not clear what is the detection limit for the rare earth elements in the more complex matrix. The hope is that you can define this in a next article.

Response: This precisely has been mentioned in our conclusions section.

Reviewer 4 Report

In the manuscript named: “Micro- Laser-Induced Breakdown Spectroscopy: A Novel approach used in the detection of Six Rare Earths and One Transition Metal“,  a micro- Laser-Induced Breakdown Spectroscopy (LIBS) instrument, housed in a glove box, has been employed to demonstrate that rare earth elements (REE) such as europium, gadolinium, lanthanum, neodymium, praseodymium, samarium and a transition metal, yttrium, can be identified individually in a complex mixture (where 5% wt of each element was mixed in a graphite matrix). These elements are important in nuclear fission reactions especially for estimation of actinide masses for non-proliferation “safeguards”. 

The authors work is modest, on the contrary the idea of the experimental set up adaptation is innovative but in my opinion is not enough to make the paper scientifically sound. The manuscript presents few results limited to the qualitative detection and identification of the emission lines from the REE elements considered.

My concern about the manuscript are itemized as follow:

1) Abstract

The authors presented in this article only data for the 5% mixture of the REEs with graphite powder along with the transition metal. Thus, there is no need to say that "...each element was mixed in the graphite matrix in different percentages from 1% to 50% by weight" because they reported data only for the 5% mixture.

2) Introduction

I will suggest to rewrite more critically the introduction section discussing a bit more in detail the previous work published on the topic. Pay also attention to quote all the existing published and recent literature.

3) Results and discussion

Since the main problem to quantify REEs by LIBS is their rich spectra and the consequent frequent spectral interferences with the matrix elements, I think is fundamental for this type of work to demonstrate and assist spectroscopists in the estimation of LIBS sensitivity. 

4) Conclusions

As the same authors mention as a future work, I think is fundamental to estimate at this stage the sensitivity in terms of the limit of detection (LOD) in order to enrich the paper.

Author Response

Thanks to Reviewer #4.

1) Abstract

The authors presented in this article only data for the 5% mixture of the REEs with graphite powder along with the transition metal. Thus, there is no need to say that "...each element was mixed in the graphite matrix in different percentages from 1% to 50% by weight" because they reported data only for the 5% mixture.

Response: The authors would like to explain that each element was mixed in the graphite matrix in different percentages from 1% to 50% by weight and the LIBS spectra were obtained for each composition as well as after mixing each element in the same amount using oxides of the elements. This was done to see what concentration was the lowest amount that can be detected using the microscopic LIBS technique. So we have collected data on all the concentrations that were prepared.

2) Introduction

I will suggest to rewrite more critically the introduction section discussing a bit more in detail the previous work published on the topic. Pay also attention to quote all the existing published and recent literature.

Response: Did add more to the introduction and 6 more references were added.

3) Results and discussion

Since the main problem to quantify REEs by LIBS is their rich spectra and the consequent frequent spectral interferences with the matrix elements, I think is fundamental for this type of work to demonstrate and assist spectroscopists in the estimation of LIBS sensitivity.

Response: We agree with is fully and this will be accomplished in the next experiments.

4) Conclusions

As the same authors mention as a future work, I think is fundamental to estimate at this stage the sensitivity in terms of the limit of detection (LOD) in order to enrich the paper.

Response:  This manuscript is meant to demonstrate the feasibility of the micro-LIBS technique for very small quantities of rare earth elements especially for hot cell environments.  We will definitely work on the LODs for the next publication.

Round 2

Reviewer 1 Report

The manuscript has been improved following referee suggestions

Reviewer 4 Report

The authors answered to all the points raised giving their reasons but in my opinion are not enough to improve the manuscript.